# Neuronal hyperactivity in neurons derived from individuals with gray matter heterotopia

Francesco Di Matteo[1,2,3], Rebecca Bonrath[1], Veronica Pravata [1], Hanna Schmidt[3], Ane Cristina Ayo Martin[2,3], Rossella Di Giaimo [1,3,4], Danusa Menegaz[3], Stephan Riesenberg [5], Femke M. S. de Vrij [6,7], Giuseppina Maccarrone[3], Maria Holzapfel[3], Tobias Straub [8], Steven A. Kushner [6,9], Stephen P. Robertson [10], Matthias Eder[3] ✉ & Silvia Cappello [1,3] ✉

Periventricular heterotopia (PH), a common form of gray matter heterotopia associated with developmental delay and drug-resistant seizures, poses a challenge in understanding its neurophysiological basis. Human cerebral organoids (hCOs) derived from patients with causative mutations in *FAT4* or *DCHS1* mimic PH features. However, neuronal activity in these 3D models has not yet been investigated. Here we show that silicon probe recordings reveal exaggerated spontaneous spike activity in FAT4 and DCHS1 hCOs, suggesting functional changes in neuronal networks. Transcriptome and proteome analyses identify changes in neuronal morphology and synaptic function. Furthermore, patch-clamp recordings reveal a decreased spike threshold specifically in DCHS1 neurons, likely due to increased somatic voltage-gated sodium channels. Additional analyses reveal increased morphological complexity of PH neurons and synaptic alterations contributing to hyperactivity, with rescue observed in DCHS1 neurons by wild-type *DCHS1* expression. Overall, we provide new comprehensive insights into the cellular changes underlying symptoms of gray matter heterotopia.

Cortical malformations (CMs) result from alterations of one or multiple neurodevelopmental steps, including progenitor proliferation, neuronal migration and differentiation. CMs are frequently associated with epilepsy, cognitive deficits and behavioral alterations[1], highlighting the importance of better understanding the pathophysiology of these disorders. One of the most common forms of CMs is periventricular heterotopia (PH), which is often associated with seizures and intellectual disabilities and is usually identified by ectopic neurons lining the ventricular zone[2–4]. Genetic causes include *FLNA* mutations and rare variants in several other genes such as *DCHS1*, *FAT4*, *ARFGEF2*, *MAP1B*, *NEDD4L*, *ECE2*[5–10]. Moreover, recently high-throughput analysis of patients with PH indicated an extreme genetic

[1]Division of Physiological Genomics, Biomedical Center (BMC), Faculty of Medicine, Ludwig-Maximilians-University (LMU), Munich, Germany. [2]International Max Planck Research School for Translational Psychiatry (IMPRS-TP), Munich, Germany. [3]Max Planck Institute of Psychiatry, Munich, Germany. [4]Department of Biology, University Federico II, Naples, Italy. [5]Max Planck Institute for Evolutionary Anthropology, Leipzig, Germany. [6]Department of Psychiatry, Erasmus MC University Medical Center, Rotterdam, The Netherlands. [7]ENCORE Expertise Center for Neurodevelopmental Disorders, Erasmus MC University Medical Center, Rotterdam, The Netherlands. [8]Bioinformatics Core, Biomedical Center (BMC), Faculty of Medicine, Ludwig-Maximilians-University (LMU), Munich, Germany. [9]Department of Psychiatry, Columbia University Medical Center, New York, NY, USA. [10]Department of Women's and Children's Health, University of Otago, Dunedin, New Zealand. ✉e-mail: eder@psych.mpg.de; silvia.cappello@bmc.med.lmu.de

heterogeneity, including copy-number variants[10,11]. As consequences, this heterogeneity might contribute to diversity and variability in disease phenotypes and clinical expression, making the investigation of CMs challenging[12].

We recently showed that critical features of PH can be modeled in human cerebral organoids (hCOs)[10,13,14]. Causative mutations in the protocadherin-encoding genes *FAT4* and *DCHS1* or knockdown of their expression induce changes in the morphology of neural progenitor cells (NPCs), contributing to a defective neuronal migration. Neurons were also affected in their migration dynamics and genes involved in axon guidance, synapse organization and ion channel generation were found dysregulated[13]. Interestingly, while knockdown of *Fat4* or *Dchs1* in mice increases progenitor cell proliferation[6], this is not the case in PH hCOs, highlighting a species-specific difference[13].

Inspired by these findings, we here addressed the important question of whether and how these mutations in *FAT4* and *DCHS1* impact neuronal network function. To this aim, we used iPSCs generated from control individuals and two different patients with PH, one was compound heterozygous for mutations in *FAT4* and one homozygous for mutation in *DCHS1* (Supplementary table 4). Complementary studies were performed using isogenic *FAT4* and *DCHS1* KO iPSCs, previously generated and validated [13], to control for possible differences due to the different genetic background.

## Results

### *FAT4* and *DCHS1* are critical players for the precise formation of neuronal activity

First, we generated 9 months old hCOs from two control individuals, two clinically characterized PH patients with respective *FAT4* or *DCHS1* mutations, and isogenic *FAT4* or *DCHS1* KO cell lines[6,13,15] (Fig. 1a). We chose this developmental stage as it was shown to consistently exhibit neuronal electrophysiological activity[16]. Immunostainings demonstrated that both FAT4 and DCHS1 hCOs at late developmental stages contain an increased percentage of progenitors (SOX2 +) and neurons (DCX + , NEUN+) (Fig. 1b–d), at the expense of glial cells (NFIA + , S100B +) (Fig. 1e, f) compared to control hCOs. The number of both excitatory and inhibitory neurons (EN, IN) was found increased in both PH hCOs (Fig. S1a–d), however their reciprocal proportion was not altered compared to controls (Fig. S1e).

Next, we performed silicon probe recordings in hCOs (Fig. 1g, h). We developed an experimental approach where hCOs were immobilized in a non-invasive manner, but could be rotated in the recording chamber with a holding frame to allow consecutive insertions of the same probe at different sites of the hCOs (Fig. S1f). Thereby, we were able to reliably record from spontaneously firing neurons at several locations within a particular hCO. Control experiments confirmed that neuronal firing activity increases upon elevation of the extracellular potassium concentration and vanishes in the presence of tetrodotoxin (TTX) (Fig. S1g). Intriguingly, both FAT4 and DCHS1 hCOs exhibited exaggerated spontaneous spike activity compared to control hCOs. This neuronal hyperactivity, which was found also in the isogenic FAT4 and DCHS1 KO hCOs, was less pronounced in the FAT4 than in the DCHS1 conditions and was quantified by mean firing rate and interspike interval (ISI) analyses (Fig. 1i–k, Fig. S1h–m). ISI calculations were employed to determine the proportion of high-frequency spikes (defined as consecutive spikes with an ISI < 500 ms) in the recordings. In the DCHS1 condition, both patient-derived and KO hCOs exhibited significantly heightened spike activity characterized by increased mean firing rates and a strikingly high percentage of high frequency spikes, even with lower ISI (Fig. 1k, Fig. S1m). In addition, burst analysis revealed an increase of the mean burst rate and reduced mean burst duration (Fig. 1l, m, Fig. S1i, l). In contrast, while in the FAT4 condition hCOs showed an increase in mean firing rate, burst rate and a decreased burst duration that was significantly different especially in the FAT4 KO hCOs (Fig. 1j, l, m,

Fig. S1j, i, l), the high frequency spikes, which were found higher only in FAT4 mutant hCOs (Fig. S1h), were not altered at lower ISI (Fig. 1k, Fig. S1m).

The increase in overall activity and high-frequency spikes suggests that PH hCOs may recapitulate some of the neurophysiological alterations present in PH patients with epilepsy. Intriguingly, these findings also suggest that the mechanisms underlying functional alterations might vary depending on the specific *FAT4* or *DCHS1* mutation.

### Transcriptome and proteome analysis suggest changes in neuronal morphology and synaptic function in PH hCOs

To dissect the two distinct but converging PH phenotypes and to gain deeper insight into molecular changes in FAT4 and DCHS1 neurons, we next performed proteome analysis of control and PH hCOs (Fig. 2a, Supplementary data 1). At the protein level, Pearson correlation analysis showed a positive correlation between FAT4 and DCHS1 hCOs compared to controls (0.418) (Fig. S2a), indicating genes dysregulated with the same trend compared to controls, such as the synaptic proteins SYT1 and STXBP1 or proteins involved in the neural development such as GAP43 and DPYSL5 that are upregulated in both PH hCOs. However, distinct dysregulated proteins were also detected between the two PH conditions, such as the synaptic proteins SYN1 and SLC1A2 upregulated in DCHS1 hCOs (Fig. S2b). Gene ontology analysis enrichments for common or specific biological processes and cellular components were identified for FAT4 and DCHS1 hCOs (Fig. 2b, c, Supplementary data 1). For instance, in both PH conditions, fundamental processes like generation of neurons, neuron development and differentiation, axonogenesis, as well as cellular components like synapse and neuron projection were dysregulated. However, FAT4 hCOs additionally showed dysregulated processes involved in presynaptic membrane and neurotransmitter transport, while alterations in the regulation of trans-synaptic signaling and synapse organization were specific to the DCHS1 condition (Fig. 2b, c, Supplementary data 1). Similar dysregulations were also found in the FAT4 KO and DCHS1 KO hCOs (Fig. S2e–h, Supplementary data 1).

To better highlight the neuronal alterations suggested by these findings, we next performed RNA-sequencing on NEUN+ nuclei (Fig. 2d). At the transcriptomic level, a positive Pearson correlation was found (0.433) and differences between FAT4 and DCHS1 neurons were identified, such as the voltage-gated sodium channel (VGSC) genes *SCN3A* and *SCN1A* upregulated in DCHS1 neurons (Fig. S2c, d). Gene ontology analysis revealed common enrichments in biological processes and cellular components for FAT4 and DCHS1 neurons consistently with our proteome analysis. However, FAT4 or DCHS1-specific enrichments were also similarly detected, highlighting the specific changes at the neuronal level. Indeed, neuron projection morphogenesis, axon development and guidance, somatodendritic compartment were found to be changed for FAT4 neurons, while alterations in the maintenance of intrinsic neuronal properties were specific to the DCHS1 condition (Fig. 2e, f, Supplementary data 2).

Taken together, these changes in molecular signatures provide further evidence for the notion that mutations in different genes, which lead to the same clinical expression, can mechanistically converge and/or diverge at different levels[12]. Moreover, these results let us speculate that DCHS1 neurons, but not FAT4 neurons, exhibit firing-promoting alterations in intrinsic cell properties, while changes in the spatial cellular organization and interaction, resulting from neuronal morphological and/or synaptic alterations, play a predominant causal role in the FAT4 condition.

### *DCHS1* mutant and KO neurons are more excitable and show a higher density of SCN3A

To test this hypothesis, we first performed patch-clamp recordings in 2D cultures (Fig. 3a). Both PH 2D cultures have similar cellular

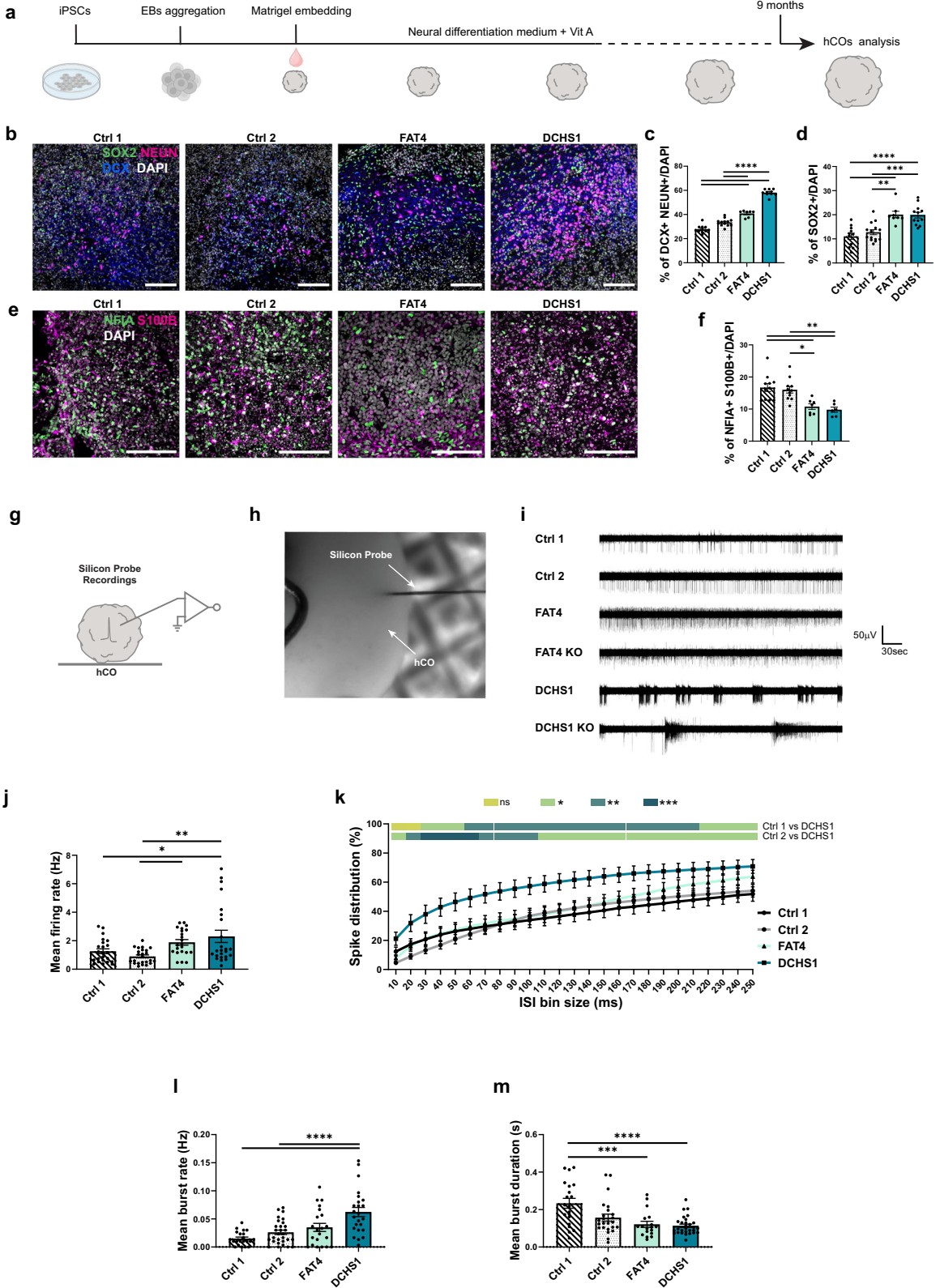

composition compared to PH hCOs, with higher percentage of progenitors (SOX2 +) and neurons (DCX +, NEUN +) (Fig. S3a–d), and lower percentage of glial cells (NFIA +, S100B +) (Fig. S3e, f) compared to controls. Furthermore, the increased percentage of excitatory and inhibitory neurons (EN, IN) was also found in both 2D PH conditions (Fig. S3g–j), while their proportion was not altered compared to controls (Fig. S3k).

Control as well as FAT4 and DCHS1 mutant and KO neurons fired solid spike trains upon depolarizing current injections (Fig. 3b, Fig. S3l). Analysis of intrinsic electrophysiological properties revealed no differences between control and FAT4 mutant and KO neurons (Supplementary Tables 1 and 2). In contrast, DCHS1 mutant and KO neurons displayed a lower action potential (AP) threshold, a higher AP overshoot and a shorter AP half-width if compared to either control

**Fig. 1 | Characterization of control-, patient-derived and KO hCOs. a** Scheme of the generation of hCOs indicating the timepoint used for immunohistochemistry and electrophysiology analysis. iPSCs (induced pluripotent stem cells), EBs (Embrio bodies), Vit A (Vitamin A), hCOs (human cerebral organoids). Micrographs of 9 months old hCOs sections immunostained for neuronal (DCX, NEUN), progenitor (SOX2) (**b**) and glial markers (NFIA, S100B) (**e**) and quantifications of the percentage of positive cells/DAPI (**c, d, f**). **g** Scheme of extracellular silicon probe recordings in an intact hCO. **h** Micrographs of a 9 months old hCO during silicon probe extracellular recording. **i** Representative recording traces of spontaneous spike activity recorded in control-, patient-derived and KO hCOs. Recordings were performed for 5 min. Quantification of the mean firing rate (**j**), spike distribution (**k**), mean burst rate (**l**) and mean burst duration (**m**) recorded in control- and patient-derived hCOs. Scale bars: 50 μm. Data are represented as mean ± SEM. Statistical significance was based on one-way (**c, d, f, j, l, m**) and two-way (**k**) ANOVA with Turkey's multiple comparison tests (*$P < 0.05$, **$P < 0.01$, ***$P < 0.001$, ****$P < 0.0001$). Independent hCOs were analyzed (**c, d, f, j–m**). Every dot in the plots refers to independently analyzed recording areas (**j, l, m**) or independent field of view (**c, d, f**). At least six ($n = 6$) randomly chosen fields of view or eighteen ($n = 18$) recording areas were analyzed across three independent batches ($N = 3$). Source data are provided as a Source Data file, including the exact *p*-values and n numbers. *Created in BioRender. Di Matteo, F. (2025)* https://BioRender.com/n12n204.

lines (Fig. 3c–f, Fig. S3m–o, Supplementary Tables 2 and 3). A possible explanation for these findings is that DCHS1 neurons possess a higher density of somatic VGSCs, as suggested from our transcriptome analysis, where different VGSC genes resulted upregulated in the DCHS1 mutant neurons (Fig. S2d). Using immunostaining, we tested this possibility for Na$_v$1.3, which is encoded by *SCN3A*. Na$_v$1.3 belongs to the most highly expressed VGSCs in the human brain and has been associated with epilepsy, intellectual disabilities and CMs[17–19]. Somatic SCN3A signal in DCHS1 mutant and KO neurons was indeed enhanced compared to FAT4 and control neurons (Fig. 3g, h, Fig. S3p, q). This increase in somatic SCN3A was also confirmed in DCHS1 hCOs (Fig. 3i, j). Interestingly, we also tested this possibility for Na$_v$1.6, another VGSC isoform associated with epilepsy[20,21] which is encoded by *SCN8A*. While somatic SCN8A signal was increased in DCHS1 mutant neurons, no differences of its levels were found between DCHS1 KO, FAT4 KO and control (Fig. S3r, s). As PH patients often present with drug-resistant epileptic seizures[22,23], using silicon probe recordings, we additionally investigated in DCHS1 hCOs whether the antiepileptic drug and use-dependent VGSC inhibitor lamotrigine[24] would dampen AP firing in human neurons derived from PH patients (Fig. 3k). Lamotrigine effectively reduced spontaneous spike and burst rate in a dose-dependent manner (Fig. 3l–n).

### FAT4 and DCHS1 are key regulators of neuronal morphology and complexity

Next, we infected all four 2D cell cultures (Ctrl 1 and 2, FAT4, DCHS1) with an adeno-associated viral vector (AAV1/2-CMV-eGFP) to sparsely label neurons and, afterwards, performed single-cell morphological reconstructions (Fig. 4a) at the same timepoint used for functional analysis. The morphology of both FAT4 and DCHS1 mutant neurons was altered (Fig. 4b, Fig. S4a–e) and Sholl analysis demonstrated a higher morphological complexity of these cells (Fig. 4c, Fig. S4f). Interestingly, the latter phenotype was particularly pronounced for FAT4 neurons (Fig. 4c). We conducted single-cell morphological reconstruction also in 2D isogenic cell cultures (Ctrl 3, FAT4 KO, DCHS1 KO) and confirmed that the altered neuronal morphology and complexity were not the result of different genetic backgrounds (Fig. S4g-n). Further analysis in 3D hCOs showed similar alterations (Fig. 4d–f, Fig. S4o–s).

To investigate the role of *FAT4* and *DCHS1* as key regulators of neuronal morphology and complexity, we acutely downregulated *FAT4* or *DCHS1* in control NPCs by transfecting previously validated miRNAs targeting these two genes[13]. Neuronal differentiation was induced 2 days post-transfection (dpt) and morphological reconstructions were conducted on neurons that differentiated for 7 days (Fig. S4t, u). FAT4 and DCHS1 knockdown neurons displayed changes in their morphology and complexity (Fig. S4v–ab), demonstrating that *FAT4* and *DCHS1* are modulators of the morphology and complexity of human neurons. Importantly, these morphological alterations were almost entirely rescued by acute overexpression of wild-type (wt) *DCHS1* in DCHS1 neurons (Fig. 4g–j, Fig. S4ac–ag).

All together, these findings show a key role of *FAT4* and *DCHS1* in the regulation of morphology and complexity of human neurons.

### FAT4 and DCHS1 regulate synaptic function

Neuronal morphology plays a prominent role in synaptic connectivity[25]. Our previous single-cell RNA-sequencing analysis performed on FAT4 and DCHS1 hCOs revealed that several genes involved in axon guidance and synapse organization were dysregulated already at early stages of development[13]. Moreover, our proteome analysis showed an upregulation of synaptic proteins in both PH hCOs, including SYN1 whose levels were even higher in the DCSH1 condition compared to FAT4 (Fig. S2a, b).

Therefore, we tested for synaptic alterations in 2D cell cultures and 3D hCOs and focused on presynaptic terminals. A significant increase in SYNAPSIN1 and SYNAPSIN2 (SYN1/2) puncta was found in the processes of both FAT4 and DCHS1 mutant and KO neurons (Fig. 5a, b, Fig. S5a, b), as well as in both FAT4 and DCHS1 mutant and KO hCOs (Fig. 5c, d, Fig. S5c, d).

Next, to investigate synaptic alterations at the functional level, we recorded miniature excitatory postsynaptic currents (mEPSCs) in 2D cell cultures and detected a significant increase of the frequency and amplitude in both FAT4 and DCHS1 neurons compared to controls (Fig. 5e–g).

As exemplarily done for the DCHS1 condition, we also tested for synaptic alterations by performing proteome analysis on fractions enriched in synaptosomes obtained from hCOs (Fig. 5h). Synaptosomes represent isolated nerve terminals and are widely used as a model system to study synaptic structure and physiology[26,27]. Most of the dysregulated proteins involved in fundamental biological processes like synapse formation, organization, maturation and neurotransmitter release (e.g., SYN1, SYN3, SYT1) were upregulated in the PH condition (Fig. 5i, Fig. S5e, f, Supplementary data 1). At the cellular component level, an enrichment of both pre- and postsynaptic compartments was detected (Fig. 5i, Supplementary data 1). Taken together, these findings indicate an involvement of *FAT4* and *DCHS1* in synaptic functions.

## Discussion

In conclusion, we here provide evidence that distinct gene mutations can cause both common and specific molecular and cellular alterations that collectively contribute to the clinical expression of PH. This is consistent with the notion that PH result from the dysregulation of different molecular pathways[9]. Although the detailed molecular processes that are controlled by FAT4 and DCHS1 remain to be elucidated, we show that *FAT4* and *DCHS1* mutations, which cause PH, induce an increase of the percentage of both excitatory and inhibitory neurons while their proportions are not altered. We also found that these mutations affect neuronal morphology and complexity and lead to synaptic alterations, indicating a key role of FAT4 and DCHS1 in these processes. These cellular effects, rather than an imbalance of excitatory-inhibitory neuron number, are likely to be causally involved in the neuronal hyperactivity detected in both FAT4 and DCHS1 mutant and KO hCOs, representing a key finding of the present study.

FAT4 and DCHS1 are critical components of the planar-cell polarity (PCP) pathway[28]. PCP proteins play a crucial role in neuronal

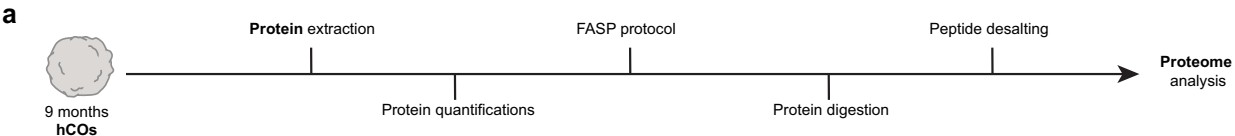

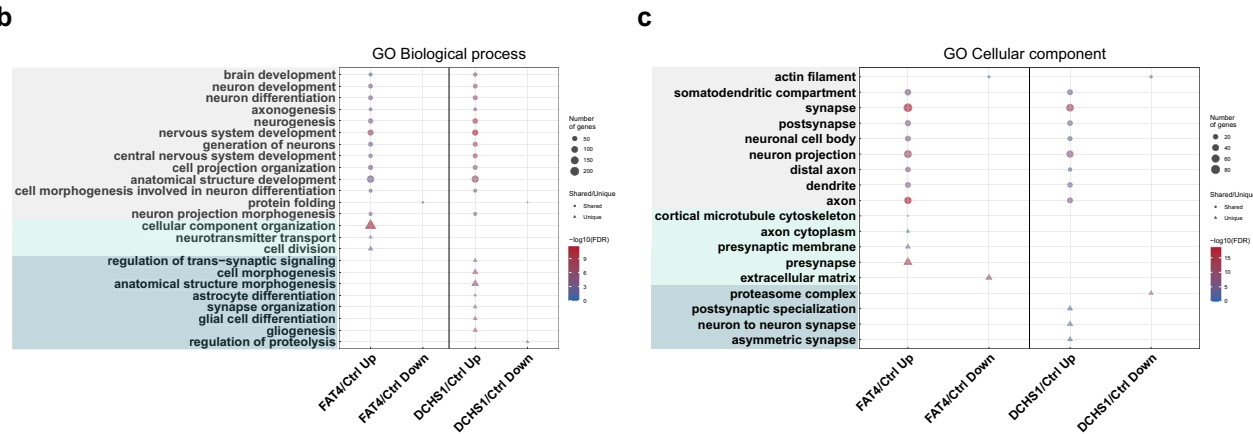

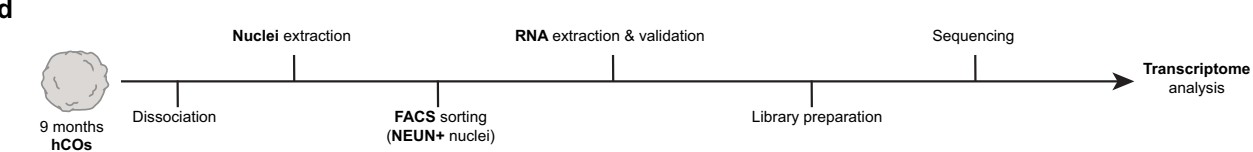

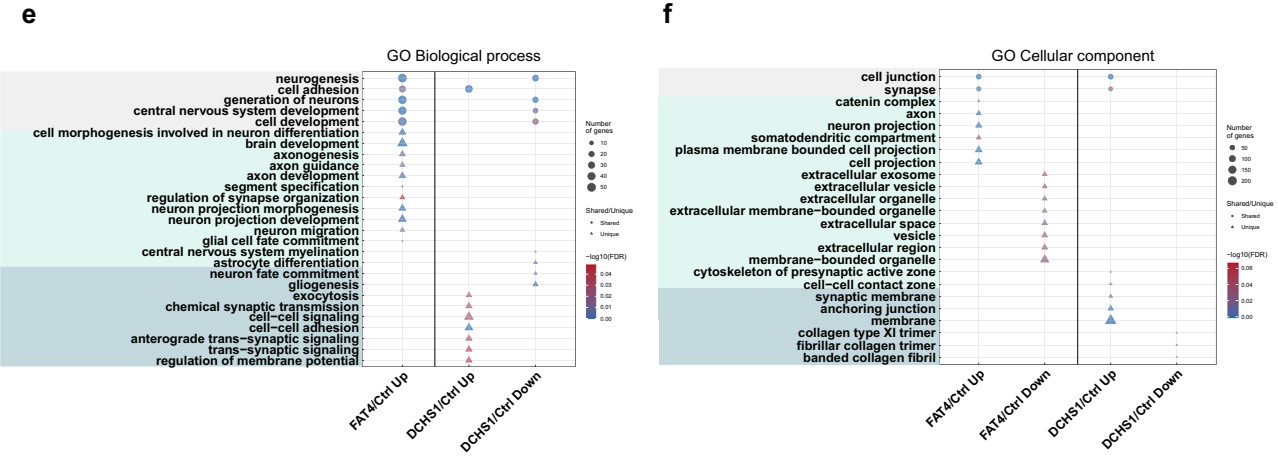

**Fig. 2 | Transcriptome and proteome analysis of control- and patient-derived and KO hCOs. a** Scheme of the samples preparation for proteome analysis. FASP (Filter Aided Sample Preparation). Biological processes (**b**) and cellular components (**c**) enriched by gene ontology (GO) analysis of the proteome data (Supplementary data 1). Common and FAT4- or DCHS1-specific enrichments are highlighted. **d** Scheme of the samples preparation for RNA-Seq analysis. NEUN+ nuclei were isolated from hCOs and processed. Biological processes (**e**) and cellular components (**f**) enriched by GO analysis of the RNA-Seq data (Supplementary data 2). Common and FAT4- or DCHS1-specific enrichments are highlighted. Statistical significance was based on Fisher's Exact.

connectivity by regulating axon guidance, dendritic arborization as well as neuronal migration and polarization[29]. Moreover, neuronal polarization is essential for the asymmetric distribution of proteins between the pre- and postsynaptic compartments[30]. Mutations in several presynaptic and postsynaptic proteins have been linked to

epilepsy and small changes in synaptic gain have been demonstrated to lead to seizure-like activity[31,32]. In particular, mutations in the gene encoding SYN1, that we found upregulated in both FAT4 and DCHS1 mutant and KO neurons and hCOs, have been associated with intellectual disability and epilepsy[33–35].

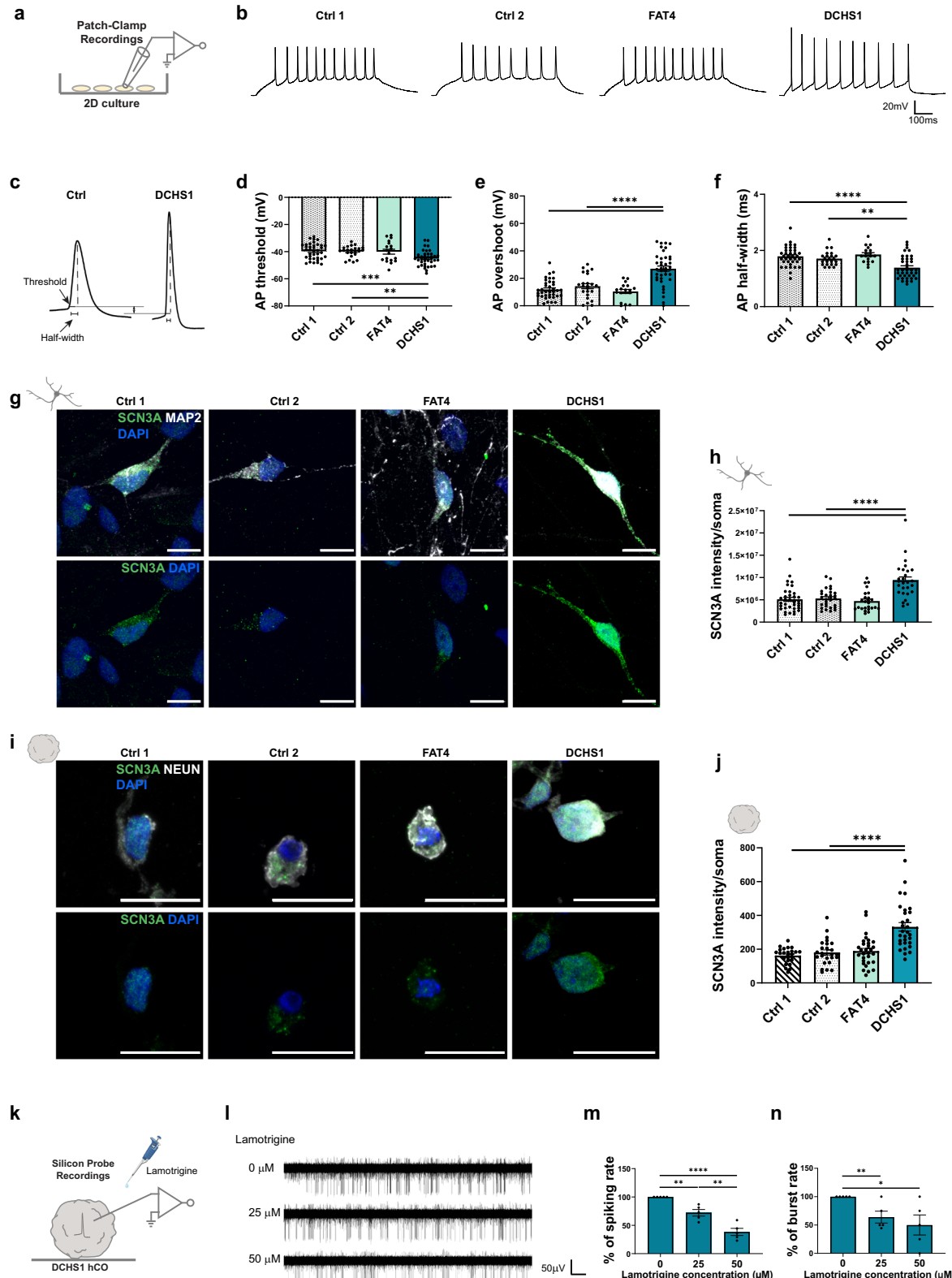

In contrast, we found alterations in intrinsic electrophysiological properties exclusively for DCHS1 neurons. These changes presumably arise from an increased density of the somatic VGSCs and the firing-promoting decrease in AP threshold is likewise compatible with the neuronal hyperactivity seen in hCOs. Despite other VGSC isoforms besides SCN3A can be altered in DCHS1 neurons, as suggested by our transcriptomic analysis, this finding aligns with the proposal that both gain- and loss-of-function SCN3A mutations may lead to increased seizure susceptibility[17]. Indeed, some pathogenic variants in SCN3A, lead to abnormal brain development and polymicrogyria (PMG) and overexpression of SCN3A in cultured human fetal neurons produced a modest increase in neurite branching without an overall increase in neurite length[20]. Notably, there were also reported infantile epilepsy cases with SCN3A variants not showing PMG[19,36]. These data suggest

**Fig. 3 | Investigation of single-cell electrophysiological properties and somatic VGSC densities in control-, patient-derived and KO neuronal cultures.** Scheme of whole-cell current-clamp recording in 10 weeks old 2D cell culture (**a**) and representative recording traces depicting evoked neuronal firing (**b**).
**c** Representative APs recorded from a control and a DCHS1 neuron. Quantification of the AP threshold (**d**), AP overshoot (**e**) and AP half-width (**f**) for control- and patient-derived neurons. **g** Micrographs of 10 weeks old control- and patient-derived 2D neurons immunostained for NEUN and SCN3A. **h** Quantification of the somatic SCN3A intensity of control- and patient-derived neurons. **i** Micrographs of 9 months old control- and patient-derived 3D neurons immunostained for NEUN and SCN3A. **j** Quantification of the somatic SCN3A intensity of control- and patient-derived 3D neurons. **k** Scheme of silicon probe recording in a DCHS1 hCO combined with lamotrigine treatment. **l** Representative traces of neuronal spike activity

recorded in a DCHS1 hCO in the absence and presence of lamotrigine. Recordings were performed for 5 min. Quantification of the percentage of spiking rate (**m**) and burst rate (**n**) in the absence and presence of lamotrigine. Scale bars: 10 μm (**g**) and 20 μm (**i**). Data are represented as mean ± SEM. Statistical significance was based on one-way ANOVA with Turkey's multiple comparison tests (**d**–**f**, **h**, **j**) and two-sided unpaired t test (**m**, **n**) (*$P < 0.05$, **$P < 0.01$, ***$P < 0.001$, ****$P < 0.0001$). Independent wells (**d**–**f**, **h**) or hCOs (**j**, **m**, **n**) were analyzed. Every dot in the plots refers to independently analyzed neurons (**d**–**f**, **h**, **j**) or independently analyzed recording areas (**m**, **n**). At least eighteen ($n = 18$) randomly chosen neurons or five ($n = 5$) recording areas were analyzed across three independent batches ($N = 3$). Source data are provided as a Source Data file, including the exact $p$-values and n numbers. *Created in BioRender. Di Matteo, F. (2025)* https://BioRender.com/m95d251, https://BioRender.com/e46e404.

that SCN3A is important for both, morphological and physiological aspects, during neuronal development. As discussed above, this DCHS1-specific phenotype could result from a dysregulation of a distinctive molecular pathway, which is consistent with a recent study that identified the previously unknown DCHS1-LIX1-SEPT9 (DLS) protein complex. The DLS complex is crucial for promoting filamentous actin organization to direct cell-extracellular matrix alignment and heart valve morphology[37]. Therefore, additional interactors of DCHS1, beyond the well-known interaction with FAT4, might exist.

The number of described individuals with *FAT4* and *DCHS1* mutations remains low and therefore identifying reliable clinical correlates that differentiate the two conditions remains difficult. The degree of intellectual disability and the distribution and extent of PH are broad for both conditions[15].

Overall, we provide detailed new insights into molecular, cellular and physiological alterations that are likely to contribute to the emergence of symptoms in gray matter heterotopia. Moreover, our study supports the use of cerebral organoids for the investigation of human brain development and disorders at different mechanistic levels, aiming for better common or personalized therapeutic interventions.

## Limitations of the study
A key limitation of our study is the use of hCOs to model brain development and neuronal network dynamics. While hCOs are a significant advance over traditional cell cultures, they primarily reflect early brain development stages and lack the full complexity of a mature brain, including certain cell types (e.g., microglia) and signaling elements. Indeed, the in vivo complexity is increasingly difficult to model in more mature stages. Thus, in vivo experiments are essential for further validation of our findings. Moreover, modeling very rare diseases leads to a limited access to patients and makes the investigation of specific phenotypes, for a certain genetic mutation, more challenging.

## Methods
### Genetic mutation in DCHS1 and FAT4
Details about the genetic mutations found in *FAT4* and *DCHS1* have been previously described[13] and can be found in the Supplementary Table 4. Sex and/or gender was not considered in the study design due to limited samples (rare disease).

### iPSCs culture
Human iPSCs previously reprogrammed from control fibroblasts and peripheral blood mononuclear cells (PBMCs) and from patient fibroblasts (FAT4 and DCHS1) were used to generate neural progenitor cells (NPCs), neurons and hCOs[13,38]. HPS0076 cells were obtained from the RIKEN Bioresource Center, Japan and generated according to the protocol in ref. 39. KO lines (FAT4 KO and DCHS1 KO) were generated and validated previously from the isogenic control HPS0076 line[13].

CTRL WTC11 iPSC line[40] was provided by Bruce R. Conklin (The Gladstone Institutes and UCSF) and NPCs were generated from the Steven Kushner lab according to the protocol in ref. 41. Details about the cell lines used in this study can be found in the Supplementary Table 6.

iPSCs were regularly checked by karyotyping and for mycoplasma regularly and were maintained on Matrigel® Basement Membrane Matrix (Corning, 354234) coated plates (Thermo Fisher), in mTESR1 medium supplemented with 1x mTESR1 supplement (Stem Cell Technologies, 85850) at 37 °C, 5% $CO_2$ and ambient oxygen level. The medium change was performed every day. For passaging, cells were treated with Accutase® solution (Life Technologies, A6964) diluted 1:4 in DPBS (Thermo Fisher, 14190144) at 37 °C for 5 min. Detached colonies were centrifuged at 300 g for 5 min to collect the pellet and resuspended in mTESR1 with 1x mTESR1 supplement and 10 μM Rock inhibitor Y-27632 (Stem Cell Technologies, 72304) and diluted as needed to the desired density.

### Generation of neurons
Controls and patients NPCs generated previously[13,41] were differentiated into neuronal cultures following the protocol described in ref. 41, with small modifications. In short, NPCs were plated on Poly-L-Ornithine (undiluted) (Sigma Aldrich, P4957)/Laminin (50 μg/ml) (Sigma Aldrich, L2020) coated 24-well plates (Thermo Fisher) in complete neural differentiation medium consisting in: Neurobasal medium (Life Technologies, 21103049), 1% N2 supplement (Life Technologies, 17502048), 2%B27-RA supplement (Life Technologies, 12587-010), 1% minimum essential medium/non-essential aminoacid (Gibco, 11140-035), 20 ng ml−1 brain-derived neurotrophic factor (PeproTech, 450-02), 20 ng ml−1 glial cell-derived neurotrophic factor (PeproTech, 450-10), 1 μM dibutyryl cyclic adenosine monophosphate (Sigma-Aldrich), 200 μM ascorbic acid (Sigma-Aldrich), 2 μg ml−1 laminin and 1% antibiotic/antimycotic (Sigma-Aldrich, A5955) and cultured for 70 days. The medium change was performed every 2–3 days. For each experiment, many independent neuronal wells (at least 3 wells) from at least three independent batches were analyzed.

### Generation and analysis of hCOs
hCOs were generated as previously described[38,42]. Briefly, iPSCs were counted and plated in a 96-well U-bottom plate at a density of 9000 live cells per 150 μL in low-bFGF hESC medium with ROCK inhibitor (1:100, final concentration 50 μM) to form Embryoid Bodies (EBs). On day 4–5, EBs were transferred to neural induction medium (DMEM-F12 with 1% (v/v) N2 supplement, 1% (v/v) GlutaMAX supplement and 1% (v/v) MEM-NEAA, heparin (final concentration of 1 μg mL⁻¹)) to promote the induction of primitive neuroepithelia. 6–7 days later, EBs were embedded in Matrigel (Corning/VWR International, 354234) droplets and kept in static culture in differentiation medium without vitamin A containing 1:1 mixture of DMEM/F12 and Neurobasal supplemented with 1:200 N2 supplement (Invitrogen), 1:100 B27 supplement without vitamin A (Invitrogen), 3.5 mL L21 2- mercaptoethanol, 1:4,000 insulin

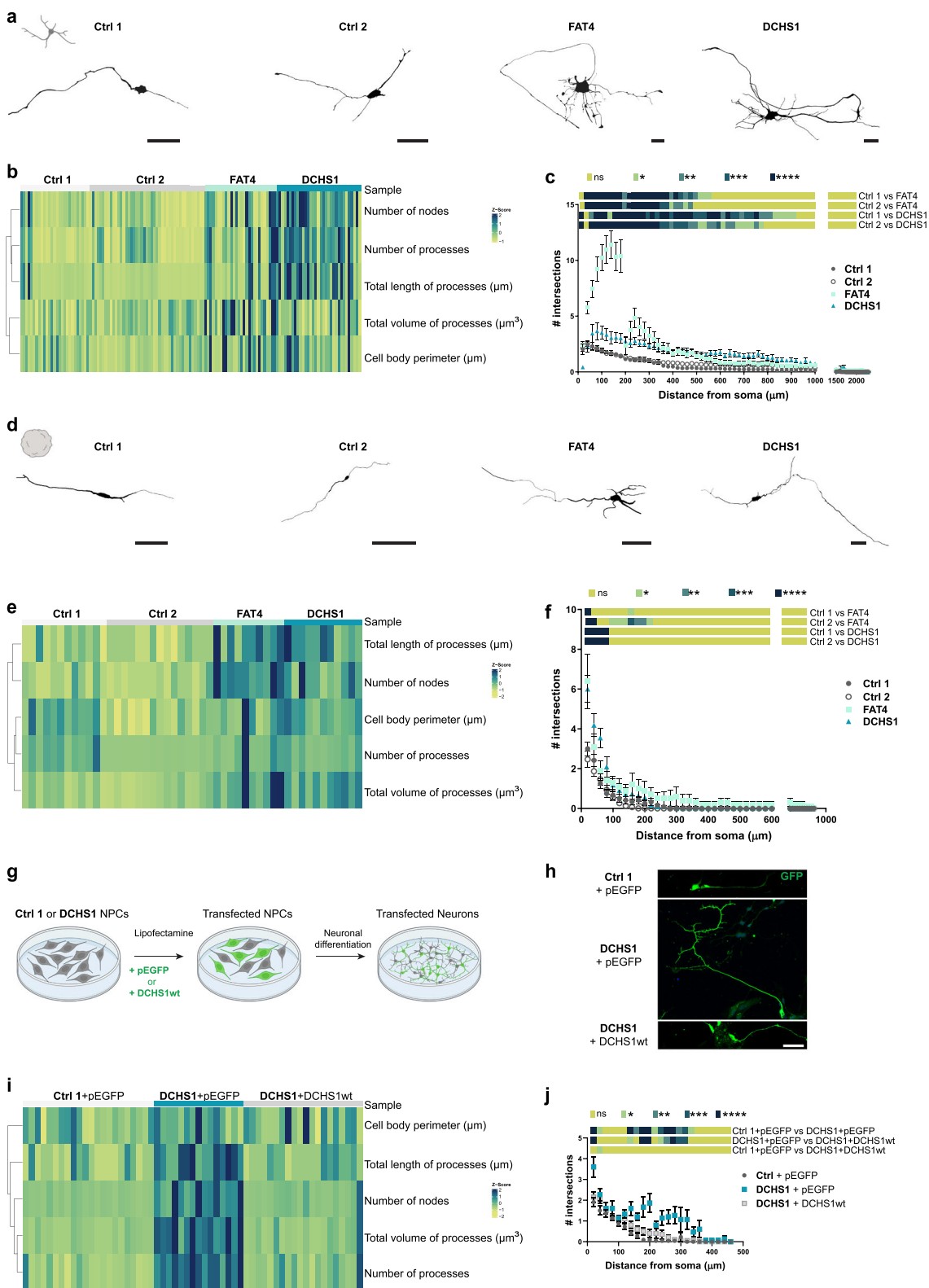

(Sigma), 1:100 Glutamax (Invitrogen), 1:200 MEM-NEAA. 4 days following the embedding, organoids were transferred to an orbital shaker in differentiation medium as above, except B27 supplement with vitamin A (Invitrogen) was used. Organoids were kept in 10-cm dishes on an orbital shaker at 37 °C, 5% CO2 and ambient oxygen level. The medium changes were performed every 3–4 days. Organoids were analyzed 9 months after the initial plating of the cells. For immunostaining, hCOs were fixed using 4% PFA for 1 h at 4 °C, cryopreserved with 30% sucrose, embedded in OCT compound mounting medium for cryotomy (VWR chemicals, 361603E) and stored at −20 °C. 16-μm sections of organoids were prepared using a cryotome. For morphological reconstructions, hCOs were embedded in 4% low melting agarose (Sigma- Aldrich, A9414) and 300 μm slices were prepared in PBS using a vibratome (speed 0,2 mm/s; amplitude 1 mm). For each experiment, different organoids for each of three different batches were analyzed.

**Fig. 4 | Morphological analysis of control- and patient-derived 2D and 3D neurons. a** Morphologies of representative 10 weeks old control- and patient-derived 2D neurons reconstructed with Neurolucida software. **b** Heatmap of the morphological characterization of reconstructed 2D neurons. Z-scores of analyzed parameters are displayed as colors ranging from yellow to blue as shown in the key. **c** Quantification of the number of intersections related to the distance from the soma of control- and patient-derived 2D neurons, obtained by Sholl analysis. **d** Morphologies of representative 9 months old control- and patient-derived 3D neurons reconstructed with Neurolucida software. **e** Heatmap of the morphological characterization of reconstructed 3D neurons. Z-scores of analyzed parameters are displayed as colors ranging from yellow to blue as shown in the key. **f** Quantification of the number of intersections related to the distance from the soma of control- and patient-derived 3D neurons, obtained by Sholl analysis. **g** Scheme of the 2D neuronal differentiation of Ctrl 1 or DCHS1 neural progenitor cells (NPCs) upon expression of GFP (pEGFP) or *DCHS1*-wt. **h** Micrographs of

sections of 7 days old control and DCHS1 neurons (9dpt) immunostained for GFP. **i** Heatmap of the morphological characterization of control and DCHS1 neurons upon *DCHS1*-wt expression. pEGFP vector was used as control. Z-scores of analyzed parameters are displayed as colors ranging from yellow to blue as shown in the key. **j** Quantification of the number of intersections related to the distance from the soma of control and DCHS1 neurons (9dpt), obtained by Sholl analysis. Scale bars: 100 μm (**a**, **d**), 50 μm (**h**). Data are represented as mean ± SEM. Statistical significance was based on two-way ANOVA with Turkey's multiple comparison tests (**c**, **f**, **j**) and Fisher's Exact (**b**, **e**, **i**) (*$P < 0.05$, **$P < 0.01$, ***$P < 0.001$, ****$P < 0.0001$). Independent wells (**c**, **j**) or hCOs (**f**) were analyzed. At least ten ($n = 10$) randomly chosen neurons were analyzed across three independent batches ($N = 3$). Source data are provided as a Source Data file, including the exact *p*-values and n numbers. *Created in BioRender. Di Matteo, F. (2025)* https://BioRender.com/m95d251, https://BioRender.com/v53q942.

## Immunohistochemistry

Immunostainings were performed as stated before[38]. In short, organoids sections and neuronal cell cultures were permeabilized with 0.3% Triton for 5 min and were then blocked with 0.1% Tween, 10% Normal Goat Serum (Biozol, VEC-S-1000). Primary and secondary antibodies were diluted in blocking solution. Nuclei were visualized using 0.5 mg/ml 4,6-diamidino-2-phenylindole (DAPI) (Sigma- Aldrich, D9542). Finally, after several washes with 0.01% Tween in 1X PBS, samples were mounted with Aqua-Poly/Mount (PolyScience, 18606-20) and left to dry in the dark before imaging.

Immunostained sections and neuronal cultures were analyzed using a Leica SP8 confocal laser-scanning microscope with 20X, 25X and 40X objectives. Z-projections were taken to obtain the full 3D image. Notably, for organoids sections a post-fixation step in 4% PFA for 10 min was performed and, for the exposure of nuclei antigen, an extra pretreatment for antigen retrieval before the post-fixation step was performed: The sections were incubated in a freshly made 10 mM citric buffer (pH 6) for 1 min at 720 W and for 10 min at 120 W and then left them to cool down for 20 min at RT. Cells quantifications were performed with the ImageJ software and analyzed with GraphPad. Antibodies list is included in the Supplementary Table 5.

## FACS analysis: nuclei isolation

Nuclei from hCOs were isolated following the protocol from[43] with small modifications[38]. For each condition, three samples were analyzed as biological replicates; every sample contained three individual hCOs. Briefly, hCOs were dissociated by Dounce homogenization and filtered with BD Falcon tubes with a cell strainer cap (Corning, 352235) to get single cells. RNase inhibitor (NEB, M0314) (0.4 U/μl) and DNase I (NEB, M0303) (1 U/μl) were added to the homogenization buffer. RNase inhibitor (0.2 U/μl) was added to the blocking solution. Primary antibody against NeuN (Millipore, MAB377) was used at 1:1500 dilution. Secondary antibody Alexa Fluor 546 Goat anti-Mouse IgG1 (c1) (Thermo Scientific, A-21123) was used at 1:2500 dilution. DAPI (D9542) (0.5 mg/ml) was added during the last 10 min of the secondary antibody incubation. FACS analysis was performed with a FACS Melody TM cell sorter (BD) in BD FACS Flow TM medium, with a nozzle diameter of 100 lm. Debris and aggregated cells were gated out by forward scatter, sideward scatter; single cells were gated out by FSC-W/FSC-A. Gating for fluorophores was done using samples stained with secondary antibody only. Sorted cells were collected in DPBS containing RNAse inhibitor (0.2 units/μl) and 1 ml QIAzol® Lysis Reagent (Qiagen, Hilden, Germany) was added to the mix and kept in −80 °C until further analysis.

## RNA extraction, cDNA synthesis, and real-time qPCR

RNA extraction, cDNA synthesis and real-time qPCR of NEUN+ and PAX6+ nuclei from FACS sorting were performed as described before[10]. Briefly, RNA was extracted using RNA Clean & Concentrator Kit (Zymo Research, R1015) and cDNA was synthesized using

SuperScript III reverse transcriptase (ThermoFisher, 18080-044) with Random primers (Invitrogen, 48190-11) according to the manufacturer's protocol. Subsequently, qPCR was performed in triplicates on a LightCycler 480 II (Roche) using the LightCycler 480 SYBR Green I Master (#04707516001, Roche).

The primers sequences were as follows:

GAPDH: FW: 5'-AATCCCATCACCATCTTCCAGGA-3'; RV: 5'-TGGA CTCCACGACGTACTCAG-3' PAX6: FW: 5'-ACCCATTATCCAGATGTGTT TGC-3'; RV: 5'-ATGGTGAAGCTGGGCATAGG-3'[44] NEUN: FW: 5'-CCAAG CGGCTACACGTCT-3'; RV: 5'-GCTCGGTCAGCATCTGAG-3'[45] NESTIN: FW: 5'-GGGAAGAGGTGATGGAACCA-3'; RV: 5'-AAGCCCTGAACCCTCT TTGC-3'[45]. Relative expression was calculated using the DDCp method.

## cDNA amplification library preparation and data analysis

All the steps for the library preparation from the RNA extracted from the NEUN+ were done according to[46]. First, the integrity of the extracted RNA was analyzed at the Bioanalyzer by using the RNA 6000 pico kit (Agilent) following the manufacturer's protocol.

First-Strand cDNA synthesis and cDNA amplification were done with the SMART-Seq® v4 Ultra® Low Input RNA Kit for Sequencing (Takara Bio USA) following the manufacturer´s protocol. For the cDNA amplification, we performed a range of PCR cycles. The amplified cDNA was purified with AMPure XP beads (Beckman Coulter) following the manufacturer's protocol. The purified sample was collected and its concentration was measured with the Qubit DNA HS kit following the manufacturer´s protocol. The concentration was ranging between 0.5 and 2 ng/μl.

The cDNA shearing was performed using the S220 Covaris with AFA technology (Covaris) and the library was prepared with Microplex Library preparation kit v2 (Diagenode) following the manufacturer's protocol. After the Covaris treatment, the resulting cDNA was in 200–500 bp size range. The concentration, measured with the Qubit DNA HS kit, was 0,5–2 ng/ul (10–20 ng in total). After the library amplification steps with the Microplex Library preparation kit v2 (Diagenode) the DNA concentration was measured again with the Qubit DNA HS Kit. The concentration of the DNA was between 5 and 6 ng/μl. The library was purified using the AMPure XP beads (Beckman Coulter) and the concentration was checked with Qubit HS DNA kit again, revealing a final concentration between 2 and 6 ng. Finally, the quality and molarity of the library were also analyzed with the DNA 1000 kit (Agilent). The prepared libraries were sequenced at LAFUGA Genomics, Genzentrum, Universität München (LMU).

## RNAseq analysis

Sequencing reads were aligned to the human reference genome (version GRCH38.100) with STAR (version 2.7.3a). Expression values (TPM) were calculated with RSEM (version 1.3.3). Post-processing was performed in R/bioconductor (version 4.0.3) using default parameters if not indicated otherwise. Differential gene expression analysis was

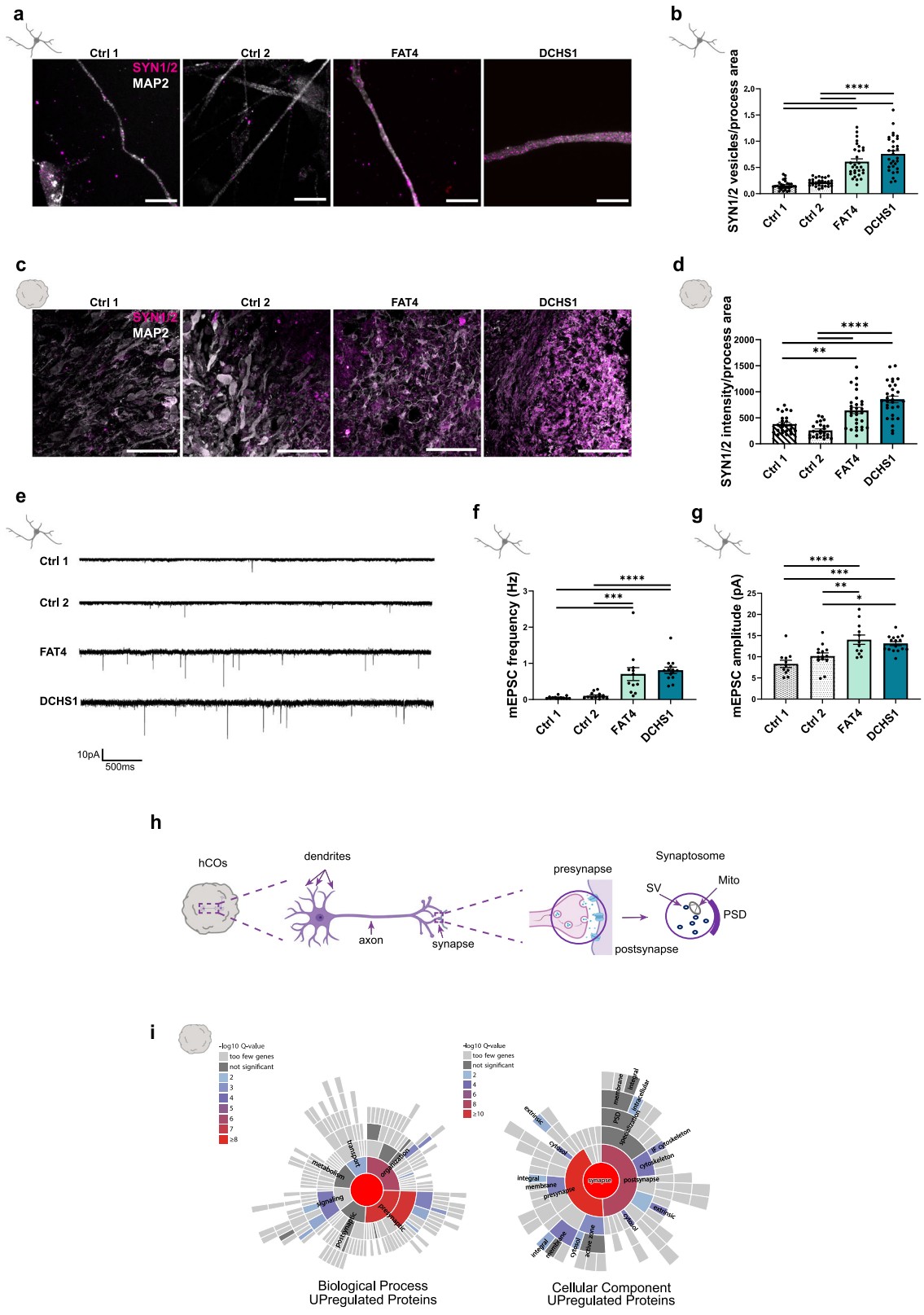

performed with 'DEseq2' (version 1.28.1). An adjusted $p$ value (FDR) of less than 0.05 was used to classify significantly changed expression.

**Silicon probe recordings in hCOs**

For recordings of spontaneous spike activity in hCOs, a particular hCO was glued with a tiny drop (0.25 µl) of Histoacryl (B.Braun) on a polypropylene mesh that was mounted on a circular plastic frame. Afterwards, the hCO was incubated at 37 °C for at least 30 min in carbogen gas (95% $O_2$/5% $CO_2$)-saturated ACSF consisting of (in mM): 121 NaCl, 4.2 KCl, 29 NaHCO$_3$, 0.45 NaH$_2$PO$_4$, 20 Glucose, 0.5 Na$_2$HPO$_4$, 1.1 CaCl$_2$ and 1 MgSO$_4$. Subsequently, the hCO was transferred with the holding device to the recording chamber and superfused with warm

**Fig. 5 | Investigation of synaptic properties in control- and patient-derived neuronal cultures. a** Micrographs of 10 weeks old control- and patient-derived 2D neurons immunostained for MAP2 and SYN1/2. **b** Quantification of the number of SYN1/2 puncta related to the MAP2 process area of control- and patient-derived neurons. **c** Micrographs of sections of 9 months old control- and patient-derived hCOs immunostained for MAP2 and SYN1/2. **d** Quantification of the intensity of SYN1/2 puncta related to the MAP2 process area of control- and patient-derived neurons. **e** Representative mEPSC recording traces obtained by whole-cell voltage-clamp measurements. Quantification of mEPSC frequency (**f**) and amplitude (**g**). **h** Scheme of isolation of synaptosomes enriched fractions from hCOs. SV (synaptic vesicles), Mito (mitochondria), PSD (postsynaptic density). **i** Graph showing significantly enriched GO terms of the proteome analysis performed on tissue isolated from hCOs, fractions enriched in synaptosomes. GO analyses show enrichment for biological process and cellular component of proteins upregulated in DCHS1 synaptosomes (Supplementary data 1). Scale bars: 10 μm (**a**), 100 μm (**c**). Data are represented as mean ± SEM. Statistical significance was based on one-way ANOVA with Turkey's multiple comparison tests (**b, d, f, g**) and Fisher's Exact (**i**) (*$P < 0.05$, ***$P < 0.001$, ****$P < 0.0001$). Independent wells (**b, f, g**) or hCOs (**d**) were analyzed. Every dot in the plots refers to independently analyzed neuronal processes (**b, d**) or neurons (**f, g**). At least twentyfive ($n = 25$) randomly chosen neuronal processes or eleven ($n = 11$) neurons were analyzed across three ($N = 3$) (**b, d**) or two ($N = 2$) (**f, g**) independent batches. Source data are provided as a Source Data file, including the exact $p$ values and n numbers. *Created in BioRender. Di Matteo, F. (2025)* https://BioRender.com/m95d251; https://BioRender.com/y48a635.

(37 °C) carbogenated ACSF at a flow rate of 2.5 ml/min. For additional stabilization, a custom-made anchor was gently placed on the top of the hCO. Recordings were performed using 16-channel probes (Cambridge Neurotech, ASSY-1 E1), which were connected to a ME2100 system (Multichannel Systems). The headstage was mounted on a PatchStar micromanipulator (Scientifica). Recording data were high-pass filtered at 100 Hz, low-pass filtered at 4 kHz, digitized at 20 kHz and transferred via a MCS IFB interface board (Multichannel Systems) to a personal computer. Data were stored using the software Multichannel Experimenter (Multichannel Systems). Visually guided insertion of the silicon probe into hCOs was conducted using a SliceScope microscope (Scientifica) equipped with a 2.5x objective. Once neuronal activity has been detected, the probe was allowed to stabilize in the tissue for 5 min and afterwards activity was recorded for 5 min. For each condition, independent recording areas were used as biological replicates. Recordings were performed in at least three independent areas per hCO, keeping a distance of at least 400 μm between two adjacent areas. Independent hCOs for each of at least three different batches were analyzed. Sample size: Ctrl 1 $n = 27$; Ctrl 2 $n = 29$; FAT4 $n = 25$; DCHS1 $n = 29$; FAT4 KO $n = 20$; DCHS1 KO $n = 25$. Analysis was conducted with the Offline Sorter™ software (Plexon) and NeuroExplorer software. Burst analysis was conducted following the NeuroExplorer firing rate based algorithm. For experiments with Lamotrigine (Sigma-Aldrich, L3791), the drug was added directly to the ACSF and recordings were started after a waiting period of 5 min. For each recording, the channel with the highest activity was used.

## Patch-clamp recordings
Somatic whole-cell current- and voltage-clamp recordings (−70 mV holding potential, >1 GΩ seal resistance, <20 MΩ series resistance, 8 mV liquid junction potential correction, 3 kHz low-pass filter, 15 kHz sampling rate) in 10 weeks old iPSCs-derived neuronal cultures were conducted at room temperature (23–25 °C) using an EPC9 amplifier (HEKA). Cells were superfused (2–3 ml/min flow rate) with the same carbogenated solution as used for silicon probe recordings. This solution additionally contained NBQX (5 μM) and picrotoxin (100 μM) for current-clamp measurements or picrotoxin (100 μM) and TTX (1 μM) for AMPA receptor-mediated mEPSC recordings, respectively. For current-clamp measurements, patch pipettes (3–5 MΩ open tip resistance) were filled with a solution consisting of (in mM): 135 KMeSO$_4$, 8 NaCl, 0.3 EGTA, 10 HEPES, 2 Mg-ATP, and 0.3 Na-GTP. Current injections were used to depolarize the neuron under investigation. All AP parameters were calculated for the first AP occurring upon depolarizing current injection. Offline analysis was performed using the software FitMaster (HEKA) and Igor Pro (WaveMetrics). AP threshold was defined as the membrane potential where the slope of the AP overshoot reached for the first time a value of 10 mV/ms. For each condition, neurons of independent wells were used as biological replicates. For current-clamp recordings, at least three independent wells for each of the three different batches were analyzed. Sample size: Ctrl 1 $n = 37$; Ctrl 2 $n = 23$; Ctrl 3 $n = 22$; FAT4 $n = 18$; DCHS1 $n = 39$;

FAT4 KO $n = 24$; DCHS1 KO $n = 31$. For mEPSC voltage-clamp recordings, the internal solution contained (in mM): 125 CsCH$_3$SO$_3$, 8 NaCl, 10 HEPES, 0.5 EGTA, 4 Mg-ATP, 0.3 Na-GTP, and 20 Na$_2$-Phosphocreatine. 10 min after break-in to the cell, mEPSCs were recorded for 5 min. Offline analysis was performed using the Mini Analysis Program (version 6.0.7, Synaptosoft). For these experiments, at least three independent wells for each of the two different batches were analyzed. Sample size: Ctrl 1 $n = 11$; Ctrl 2 $n = 13$; FAT4 $n = 12$; DCHS1 $n = 14$.

## Morphological reconstruction of neurons and analysis
10-week-old neurons and acute slices of 9-month-old hCOs were used for the morphological analysis. A sparse labeling of cells by using an adeno-associated viral vector (AAV1/2-CMV-eGFP, titer: 2.48 × 10$^{12}$ gc/mL; produced at the MPIP, transfer plasmid source: Addgene 105530) was performed. 3 days after infection, neuronal cultures and hCOs were fixed at RT for 15 min and 30 min respectively, with 4% PFA and stained for GFP and MAP2 following the described immunostaining protocol. For KD and *DCHS1* expression experiments, 1-week-old neurons were used for the morphological analysis. Neurons were visualized by using a Leica SP8 confocal laser-scanning microscope with 40x objective. Z-projections with a Z-Step Size of 0.50 μm were taken to obtain 3D image. Confocal pictures were used in the Neurolucida Software (MBF Bioscience, Neurolucida version 2017.03.3, 64-bit) and the subsequent tracing was performed in 3D. For the analysis of the reconstructed neurons, Neurolucida Explorer (MBF Bioscience, Neurolucida version 2017.02.9) was used. The "Branched Structure Analysis" tool was used to perform the neuron summary analysis and the individual process analysis. Furthermore, neuronal complexity was obtained using the Sholl analysis tool. To design concentric circles a radius increment of 20 μm were selected. Data were analyzed and visualized on RStudio Complex Heatmap package[47,48]. Statistical tests were performed in GraphPad. For each condition, neurons of independent wells and slices of independent hCOs were analyzed as biological replicates. At least three independent wells for each of the three different batches and three independent hCOs were analyzed. Sample size (2D): Ctrl 1 $n = 27$; Ctrl 2 $n = 38$; Ctrl 3 $n = 30$; FAT4 $n = 27$; DCHS1 $n = 33$; FAT4 KO $n = 38$; DCHS1 KO $n = 29$; Ctrl 1+miRNA NEG $n = 17$; Ctrl 1+miRNA FAT4 $n = 18$; Ctrl 1+miRNA DCHS1 $n = 14$; Ctrl 1+pEGFP $n = 22$; DCHS1+pEGFP $n = 15$; DCHS1 + DCHS1 WT $n = 20$. (3D): Ctrl 1 $n = 12$; Ctrl 2 $n = 15$; FAT4 $n = 10$; DCHS1 $n = 11$.

## NPCs transfection
Transfection experiments were performed following the manufacturer's protocol of Lipofectamine 3000 (Invitrogen, L3000001). Briefly, to knockdown FAT4 and DCHS1, 90% confluent control NPCs were transfected with microRNA against DCHS1 and FAT4 previously generated and validated:

DCHS1: "TGCTGTACACTGTCAGGTTGATCTCCGTTTTGGCCACT GACTGACGGAGATCACTGACAG"

FAT4: "GCTGATCAGTTGCAGTAACAGAGGAGTTTTGGCCACTGA CTGACTCCTCTGTCTGCAACT"[13]. To overexpress DCHS1 wt in DCHS1

mutant NPCs, we used the plasmid constructs previously described[13]. DCHS1 wt protein sequence was fused to GFP in pEGFP-C1 plasmid, and the empty vector was used as control. In total, 1 µg of a specific microRNA and plasmid were transfected in NPCs. 2 days post-transfection, NPCs were used to start the neuronal differentiation as described above.

## Quantifications and statistical analyses

Statistical analysis and plotting of data were performed with GraphPad Prism® version 7.04. Outliers were removed with the ROUT method[49]. Statistical significance between unpaired groups was analyzed using t-test. Statistical significance for comparison involving multiple groups was analyzed using one-way ANOVA with Turkey's multiple comparison and two-way ANOVA with Turkey's multiple comparison tests to account for two variables as indicated in the figure legends. All experiments were reproduced at least three times independently and all attempts at replication were successful. No randomization was performed, but different batches of cerebral organoids for each experiment were used (at least $N = 3$). All acquired data were verified by a second investigator.

## Isolation of synaptosomal enriched fractions from hCOs

Synaptosomal fractions were prepared as previously described[27] with small changes. Briefly, hCOs were homogenized in nine volumes of cold isotonic medium (HM: 0.32 M sucrose, 10 mM Tris-HCl, pH 7.4), using a Dounce homogenizer. After centrifugation of the homogenate (2200 g, 1 min, 4 °C), the sediment was resuspended in the same volume of HM and centrifuged under the same conditions to yield a washed sediment containing nuclei, cell debris, and other particulates (P1 fraction). The two supernatant fractions were mixed and centrifuged at a higher speed (22,000 g, 4 min, 4 °C), to obtain a second sediment that was resuspended in the same volume of HM and centrifuged as described above. The washed sediment contained free mitochondria, synaptosomes, myelin, and microsomal fragments (P2 fraction). The sediment was homogenized in HM and used for subsequent analyses.

## Sample preparation for mass spectrometry

Full lysate and synaptosome enriched samples were resuspended in RIPA buffer, the protein amount was estimated by Bradford protein assay. Protein extract containing 20 µg of protein was subjected to the modified FASP protocol[50]. Briefly, the protein extract was loaded onto the centrifugal filter CO10 kDa (Merck Millipore, UFC201024), and detergent were removed by washing five times with 8 M Urea (Merck Millipore) 50 mM Tris (Sigma-Aldrich) buffer. Proteins were reduced by adding 5 mM dithiothreitol (DTT) (Bio-Rad) at 37 °C for 1 h at dark. To remove the excess of DTT, the protein sample was washed three times with 8 M Urea, 50 mM Tris. Subsequently protein thiol groups were blocked with 10 mM iodoacetamide (Sigma-Aldrich) at RT for 45 min. Before proceeding with the enzymatic digestion, urea was removed by washing the protein suspension three times with 50 mM ammonium bicarbonate (Sigma-Aldrich). Proteins were digested first by Lys-C (Promega) at RT for 2 h, then by trypsin (Premium Grade, MS Approved, SERVA) at RT, overnight, both enzymes were added at an enzyme-protein ratio of 1:50 (w/w). Peptides were recovered by centrifugation followed by two additional washes with 50 mM ammonium bicarbonate and 0.5 M NaCl (Sigma-Aldrich). The two filtrates were combined, the recovered peptides were lyophilized under vacuum. Dried tryptic peptides were desalted using C18-tips (Thermo Scientific), following the manufacture instructions. Briefly, the peptides dissolved in 2% (v/v) formic acid (Thermo scientific) were loaded onto the C18-tip and washed 10 times with 0.1% (v/v) formic acid, subsequently the peptides were eluted by 95% (v/v) acetonitrile (Merck Millipore), 0.1% (v/v) formic acid. The desalted peptides were lyophilized under vacuum. The purified peptides were reconstituted in 0.1% (v/v) formic acid for LC-MS/MS analysis.

## Mass spectrometry analysis

Desalted peptides were loaded onto an in-house pulled nano capillary (15 cm, 75 µm ID, Sutter instrument, Puller P-1000), packed with C18 (ReproSil-Pur 1.9 µm, Dr. Maisch GmbH) via the autosampler of the Ultimate 3000 R n-LC system (Dionex). Eluting peptides were directly sprayed onto the Q-Exactive Plus Mass Spectrometer coupled to the Nano Spray Flex source (Thermo Fisher). Peptides were loaded in buffer A (0.1% (v/v) formic acid at 300 nl/min and percentage of buffer B (98% acetonitril, 0.1% formic acid) the first 5 min solvent B was maintained at 2%, then it was ramped from 2% to 30% over 120 min followed by a ramp to 60% over 20 min then 98% over 1 min, and maintained at 98% for another 5 min, decreased at 2% over 1 min and maintained at 2% for 10 min. The data acquisition was performed using Xcalibur software (Thermo Fisher). The mass spectrometer was operating in data dependent acquisition mode including one FT survey scan followed by ten HCD MS/MS scans per acquisition cycle. The analysis was performed in the mass range from 375 to 1400 m/z, resolution 70,000 and AGC target 3E6, positive mode. The ionization voltage was 1.9 kV. The HCD fragmentation was performed at resolution 17,500, AGC target 1E5, normalized collision energy (N CE) 27%, max. injection time 100 ms and dynamic exclusion 30 s.

## Data analysis

Raw data were processed using the MaxQuant computational platform (version 1.6.17.0)[51] with standard settings applied for ITMS Ion trap data. Shortly, the peak list was searched against the Uniprot database of Human database (downloaded in October 2020) with an allowed precursor mass deviation of 10 ppm and an allowed fragment mass deviation of 20 ppm. MaxQuant by default enables individual peptide mass tolerances, which was used in the search. Cysteine carbamidomethylation was set as static modification, and methionine oxidation and N-terminal acetylation as variable modifications. The match-between-run option was enabled, and proteins were quantified across samples using the label-free quantification algorithm in MaxQuant generating label-free quantification (LFQ) intensities.

## Proteomic analyses

Proteomic data were processed using the RStudio package "DEP"[52] and following the LFQ-based differential analysis. The MaxQuant output table 'proteingroups.txt' was used as input and data were prepared and processed for differential analysis. Result table was then extracted, and results were plotted using RStudio packages ggplot2, dyplyr and tidyverse. An p value of less than 0.05 and a log2FoldChange >1 for upregulated proteins and <1 for downregulated proteins was used to classify significantly changed expression. For synaptosome fraction analyses, data were further analyzed using SynGO, an interactive knowledge base that accumulates available research about synapse biology using Gene Ontology (GO) annotations to novel ontology terms[53].

## GO enrichment analysis

Gene ontology (GO) enrichment analysis was conducted using the *gprofiler2* package in R[54]. Enrichment was performed for the categories of biological process (BP) and cellular component (CC) to identify terms associated with gene sets from each condition. Terms were filtered to highlight those unique to individual conditions or shared across conditions, allowing for a comparative view of distinct and overlapping pathways. Statistically significant GO terms were identified based on adjusted p-values using the Benjamini-Hochberg FDR correction.

Data visualization was carried out using *ggplot2*[55] in R to illustrate the enriched terms across conditions, with specific emphasis on unique and shared terms. This approach enabled a clear graphical representation of the differences and similarities in enriched biological processes and cellular components among the conditions.

**Reporting summary**

Further information on research design is available in the Nature Portfolio Reporting Summary linked to this article.

## Data availability

All genomic data will be available at the GEO database under accession number GSE220673. The mass spectrometry proteomics data will be available at the the ProteomeXchange Consortium via the PRIDE[56,57] partner repository with the dataset identifier PXD038760. Source data are provided with this paper.

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

## Acknowledgements

We thank the families participating in this study for their involvement. We thank Barbara Hauger for their excellent electrophysiological technical support; Rosa-Eva Huettl for help providing AAV; Filippo Cernilogar for contribution with transcriptomic data. We thank the core facility bioimaging (cf bim) of the Biomedical Center (BMC) of the Ludwig-Maximilians-Universität München for their help during the revision process. We also thank all members of the Cappello lab, for technical help, and critical discussions, in particular Marie Lackmann for help with immunostainings and Andrea Forero for help generating schematic drawings. This work was supported by funding from the Max Planck Society, ERA-Net Neuron (nEUrotalk) 01EW1907, ERA-Net E-Rare (HETER-OMICS) 01GM1914, the European Union (ERC Consolidator Grant, Exo-Devo | 101043959), Horizon-EIC–2022-pathfinderopen 3D-BrAIn – 101098791 and by the Netherlands Organ-on-Chip Initiative, an NWO Gravitation project (024.003.001) funded by the Ministry of Education, Culture and Science of the government of the Netherlands and ZonMW PSIDER program TAILORED (10250022110002) (to SAK, FMSDV). Francesco Di Matteo is supported by ERA-Net Neuron (nEUrotalk).

## Author contributions

S.C. conceived the project. S.C. and F.D.M. designed the experiments. M.E. designed the electrophysiological experiments. F.D.M., R.B., H.S., A.C.A.M., D.M., and R.D.G. performed the experiments. VP performed proteomic and transcriptomic data analysis. R.B. performed and quantified immunostainings of hCOs. D.M. analyzed EPSC recordings. T.S. helped with the transcriptomic data alignment. F.D.M., R.D.G., and A.C.A.M. prepared samples for transcriptomic analysis. V.P. and G.M. prepared samples for proteomic analysis and validated MS results. S.P.R. provided patients's fibroblasts. S.K. and F.dV. provided control NPCs. M.H. generated AAV. SR generated the FAT4 and DCHS1 KO lines. F.D.M., S.C., and M.E. wrote the manuscript. All authors provided ongoing critical review of results and commented on the manuscript.

## Funding

## Competing interests

The authors declare no competing interests.
