## [Transparent Peer Review file · Nature Communications]

Neuronal Hyperactivity in Neurons Derived from Individuals with Grey Matter Heterotopia

Corresponding Author: Professor Silvia Cappello

Version 0:

Reviewer comments:

Reviewer #1

(Remarks to the Author)

PH is a common form of grey matter heterotopia associated with developmental delays and drug-resistant seizures. In this manuscript, the authors have used human cerebral organoids derived from patients with mutations in the FAT4 or DCHS1 genes to model PH features (Klaus et al. 2019). The neuronal activity in these models had not been previously demonstrated. In this study, Di Matteo and colleagues used silicon probe recordings to investigate neuronal activity in cerebral organoids derived from patients with FAT4 or DCHS1 mutations. They found increased spontaneous spike activity indicating functional changes in neuronal networks. The data were confirmed with isogenic KO lines with mutations in the 2 genes, excluding that the phenotype observed is only caused by individual variability. Transcriptome and proteome analyses revealed changes in gene ontology terms related to neuronal morphology and synaptic function. In addition, using patch-clamp recordings they showed that neurons with DCHS1 mutations had an additional intrinsic phenotype, likely due to an increase in somatic voltage-gated sodium channels. The authors then performed morphological reconstructions and immunostainings showing that mutant neurons have both morphological and synaptic alterations, contributing to their hyperactivity. Rescue experiments were also performed.

The experiments are well-designed, and the results provide critical insights into the molecular, cellular and physiological changes underlying the symptoms of patients with grey matter heterotopia.

However, I have couple of suggestions that will improve the impact of the study:

- 1) Characterization of 2D cultures, e.g., cellular composition, is essential for assessment of their similarity to human cortical organoids. In the current study, proportions of different cell types including progenitors, astrocytes, and neurons in 2D cultures has not been demonstrated. This would validate the 2D culture strategy for intracellular characterization, patch clamp experiments and other comparative analyses.
- 2) Performing miniature excitatory postsynaptic current (mEPSC) recordings on control and patient-derived neurons could provide significant insights into synaptic alterations. These recordings could reveal synaptic dysfunctions in neurons with FAT4 and DCHS1 mutations. This would support the observed increase in SYN1/2 staining in both 2D cultures and 3D hCOs, as well as the proteomic data from synaptosomes. Confirmation of these synaptic alterations at the electrophysiological level would strengthen the evidence of synaptic pathology in the context of these genetic conditions.
- 3) Specify the timepoint used for the 2D functional and morphological analyses. The immunostainings were done on 10-week-old neurons, were all the 2D analyses done at this timepoint? If yes, please justify the choice of that timepoint?
- 4) Mutations in different SCNs isoforms have been associated with epilepsy, have any alterations in other isoforms been examined (e.g., the SCN8A or SCN1A isoform)? It could be a point worth to add in discussion.
- 5) In the discussion, the authors state 'Mutations in several presynaptic and postsynaptic proteins have been linked to epilepsy and small changes in synaptic gain have been demonstrated to lead to seizure-like activity'. Since the authors have shown dysregulations of SYN1/2 both via proteomic and immunostaining analysis in FAT4 and DCHS1 conditions, have specific alterations in SYN1/2 been associated with epilepsy?

Reviewer #2

(Remarks to the Author)

In this manuscript, Di Matteo and colleagues demonstrate neuronal hyperactivity in Grey Matter Heterotopia neurons using human cerebral organoids (hCOs) and human 2D cultures. The authors first utilized silicon probes to show increased excitability in hCOs derived from patients with FAT4 and DCHS1 mutations, controls, and isogenic KO lines. They then performed transcriptomic and proteomic analyses, identifying dysregulated genes associated with neuronal morphology and synapse function. Moving to 2D-derived human neurons, they observed a decrease in action potential threshold and morphological changes. Although the study is novel and highly relevant to the field, several points need to be addressed:

1. Isogenic KOs: The rationale for choosing these isogenic lines over isogenic controls derived from FAT4 and DCHS1 mutations is unclear. While isogenic lines are crucial for understanding the phenotype, the isogenic KO lines are absent in key experiments such as transcriptomics and proteomics (and e.g., Fig. 5d). Additionally, the isogenic KOs are linked to just a control line, necessitating consistent comparisons. Some results are weaker when comparing the control line with their isogenic KOs (e.g., Fig. 1). Please see below for statistical analysis.
2. Statistics: The statistical tests used throughout the manuscript (T-test or Mann-Whitney) are inappropriate. Proper multiple comparison tests and ANOVA should be used for results involving multiple groups. In some cases (e.g., Fig. 1k), a two-way ANOVA should be employed to account for two variables (genotype and bin size). A linear mixed model would be ideal for data from different organoids/2D cultures to mitigate variability between differentiations affecting the analysis.
3. 2D Neurons: The rationale for transitioning from hCOs to 2D cultures is unclear, especially since previous work by the same group showed migration and maturation deficits in hCO models that cannot be recapitulated in 2D systems. How do the 2D data correlate with the phenotypes observed in 3D models?
4. Omics: Only upregulated pathways are shown. What about downregulated pathways?
5. SCN3A: The choice of this sodium channel requires clarification. The current results do not convincingly demonstrate SCN3A involvement. Is the antibody against SCN3A validated? Antibodies for sodium channels are often nonspecific. Why were experiments conducted with Lamotrigine, which blocks all sodium channels? Showing a greater effect of Lamotrigine on mutant lines compared to controls would provide insights into sodium channel expression levels, even if not specific to SCN3A.
6. Discussion: The limitations of the study should be discussed in more detail.
7. Introduction: Citations 5 and 6 do not reference these gene mutations and require different citations.

Reviewer #3

(Remarks to the Author)

In this article, the authors investigate neuronal networks in periventricular heterotopia using FAT4 and DCHS1 mutant iPSC-derived cerebral organoid models. The results indicate distinct electrophysiological properties between the two PH models; DCHS1 organoids being more excitable and presenting with increased density of voltage-gated sodium channels than FAT4 organoids. The experiments were thorough and well-carried out, identifying differences between the two mutational conditions in periventricular heterotopia in diverse assays including cell type enrichment, electrophysiology, multi-omics, and neuron morphology.

An area of improvement for this manuscript is the relevance of these models, and their distinctions, to the clinical manifestations of PH. This is briefly addressed (line 116) by indicating clinical expression is the same across the two mutations. However, this is not reflected in the findings, which report notable differences in the phenotypic manifestations of these mutations in vitro. To improve impact, consider correlating findings to clinical presentations of PH.

Reviewer #4

(Remarks to the Author)

Version 1:

Reviewer comments:

Reviewer #1

(Remarks to the Author)

All my concerns were addressed in the revised manuscript.

Reviewer #2

(Remarks to the Author)

We thank the authors for their replies, we have no further comments.

Reviewer #3

(Remarks to the Author)

The authors have adequately addressed the comments. We have no further suggestions for improvements.

Reviewer #4

(Remarks to the Author)

Reviewer #1 (Remarks to the Author):

PH is a common form of grey matter heterotopia associated with developmental delays and drug-resistant seizures. In this manuscript, the authors have used human cerebral organoids derived from patients with mutations in the FAT4 or DCHS1 genes to model PH features (Klaus et al. 2019). The neuronal activity in these models had not been previously demonstrated. In this study, Di Matteo and colleagues used silicon probe recordings to investigate neuronal activity in cerebral organoids derived from patients with FAT4 or DCHS1 mutations. They found increased spontaneous spike activity indicating functional changes in neuronal networks. The data were confirmed with isogenic KO lines with mutations in the 2 genes, excluding that the phenotype observed is only caused by individual variability. Transcriptome and proteome analyses revealed changes in gene ontology terms related to neuronal morphology and synaptic function. In addition, using patch-clamp recordings they showed that neurons with DCHS1 mutations had an additional intrinsic phenotype, likely due to an increase in somatic voltage-gated sodium channels. The authors then performed morphological reconstructions and immunostainings showing that mutant neurons have both morphological and synaptic alterations, contributing to their hyperactivity. Rescue experiments were also performed.

The experiments are well-designed, and the results provide critical insights into the molecular, cellular and physiological changes underlying the symptoms of patients with grey matter heterotopia.

However, I have couple of suggestions that will improve the impact of the study:

1) Characterization of 2D cultures, e.g., cellular composition, is essential for assessment of their similarity to human cortical organoids. In the current study, proportions of different cell types including progenitors, astrocytes, and neurons in 2D cultures has not been demonstrated. This would validate the 2D culture strategy for intracellular characterization, patch clamp experiments and other comparative analyses.

We thank the reviewer for the comments, as this was an essential point to validate the 2D culture strategy we used for our deeper analysis at the single cell level, we conducted a more comprehensive and meticulous characterization through immunohistochemistry, of our 2D neuronal cultures (Extended Data Fig. 3a-k). Remarkably, both culture models consistently yielded highly congruent data.

2) Performing miniature excitatory postsynaptic current (mEPSC) recordings on control and patient-derived neurons could provide significant insights into synaptic alterations. These recordings could reveal synaptic dysfunctions in neurons with FAT4 and DCHS1 mutations. This would support the observed increase in SYN1/2 staining in both 2D cultures and 3D hCOs, as well as the proteomic data from synaptosomes. Confirmation of these synaptic alterations at the electrophysiological level would strengthen the evidence of synaptic pathology in the context of these genetic conditions.

In response to this important comment, we included recordings of miniature excitatory postsynaptic currents (mEPSCs) in 2D cell cultures and detected a significant increase of the

frequency and amplitude in both FAT4 and DCHS1 neurons compared to controls (Fig.5e-g). This finding confirmed the synaptic alterations also at the functional level of both PH conditions.

3) Specify the timepoint used for the 2D functional and morphological analyses. The immunostainings were done on 10-week-old neurons, were all the 2D analyses done at this timepoint? If yes, please justify the choice of that timepoint?

We used 10 weeks old neurons for our 2D functional, morphological and immunohistochemical analyses. As this information was not clear, we have clarified it in the manuscript.

We choose this timepoint because it was shown to consistently exhibit neuronal electrophysiological activity (Gunhanlar et al. 2018)⁴⁴

4) Mutations in different SCNs isoforms have been associated with epilepsy, have any alterations in other isoforms been examined (e.g., the SCN8A or SCN1A isoform)? It could be a point worth to add in discussion.

Following this suggestion, we investigated changes in SCN8A and we found a significant increase in this VGSC isoform in DCHS1 mutant neurons compared to control and FAT4 mutant neurons. However, investigation in the KO lines did not reveal any difference in its signal between DCHS1 KO and control or FAT4 KO neurons (Extended data Fig. 3r,s).

5) In the discussion, the authors state 'Mutations in several presynaptic and postsynaptic proteins have been linked to epilepsy and small changes in synaptic gain have been demonstrated to lead to seizure-like activity'. Since the authors have shown dysregulations of SYN1/2 both via proteomic and immunostaining analysis in FAT4 and DCHS1 conditions, have specific alterations in SYN1/2 been associated with epilepsy?

We thank the reviewer for this comment. Mutations in SYN1 have been associated with epilepsy and we have included this in the discussion.

Reviewer #2 (Remarks to the Author):

In this manuscript, Di Matteo and colleagues demonstrate neuronal hyperactivity in Grey Matter Heterotopia neurons using human cerebral organoids (hCOs) and human 2D cultures. The authors first utilized silicon probes to show increased excitability in hCOs derived from patients with FAT4 and DCHS1 mutations, controls, and isogenic KOs. They then performed transcriptomic and proteomic analyses, identifying dysregulated genes associated with neuronal morphology and synapse function. Moving to 2D-derived human neurons, they observed a decrease in action potential threshold and morphological changes. Although the study is novel and highly relevant to the field, several points need to be addressed:

1. Isogenic KOs: The rationale for choosing these isogenic lines over isogenic controls derived from FAT4 and DCHS1 mutations is unclear. While isogenic lines are crucial for understanding the phenotype, the isogenic KO lines are absent in key experiments such as transcriptomics and proteomics (and e.g., Fig. 5d). Additionally, the isogenic KOs are linked to just a control line,

necessitating consistent comparisons. Some results are weaker when comparing the control line with their isogenic KOs (e.g., Fig. 1). Please see below for statistical analysis.

We thank the reviewer for these important and relevant comments. The choice to use DCHS1 and FAT4 KO isogenic lines is due to the variability of mutations (leading to PH) in the two different genes. By using KO lines, we were able to study the general function of DCHS1 and FAT4 and clearly demonstrate the role of these two proteins in the development of the phenotypes independently of the different genetic backgrounds of the patients.

We have now included proteome analysis of isogenic FAT4 and DCHS1 KO hCOs and we found similar dysregulation in both KO lines compared to the mutant conditions (Extended data Fig.e-h). Moreover, as recommended for Fig.5d, we have also examined SYN1/2 alterations in 3D KO hCOs (Extended data Fig. 5c,d) and recapitulated the findings in line with our previously shown results.

Furthermore, to make the comparisons between the different conditions easier for the reader, we have decided to separate the mutant and KO data into different graphs and, regarding Fig.1, we have better explained in the manuscript the functional differences we found between the FAT4 or DCHS1 condition.

2. Statistics: The statistical tests used throughout the manuscript (T-test or Mann-Whitney) are inappropriate. Proper multiple comparison tests and ANOVA should be used for results involving multiple groups. In some cases (e.g., Fig. 1k), a two-way ANOVA should be employed to account for two variables (genotype and bin size). A linear mixed model would be ideal for data from different organoids/2D cultures to mitigate variability between differentiations affecting the analysis.

As recommended, we modified all statistical tests. Upon removing outliers, we used one-way ANOVA with Turkey's multiple comparison tests for comparison involving multiple groups and two-way ANOVA with Turkey's multiple comparison tests to account for two variables.

3. 2D Neurons: The rationale for transitioning from hCOs to 2D cultures is unclear, especially since previous work by the same group showed migration and maturation deficits in hCO models that cannot be recapitulated in 2D systems. How do the 2D data correlate with the phenotypes observed in 3D models?

Despite our attempts to perform patch clamp recordings in human cerebral organoids (hCOs), we encountered significant technical challenges that led us to switch to 2D culture.

However, we also performed a comprehensive comparative analysis of morphological and immunohistochemical aspects in hCOs where we found similar results for neuronal morphological characterization and SCN3A and SYN1/2 immunohistochemical characterization between 3D and 2D systems.

Moreover, we have now included substantial additional data to better ensure the validation of both model systems,. This includes a thorough examination of the cell composition in our 2D neuronal cultures (Extended Data Fig. 3a-k). Remarkably, both the 2D and 3D culture models consistently yielded highly congruent data.

4. Omics: Only upregulated pathways are shown. What about downregulated pathways?

We have now also included downregulated pathways (Fig. 2b,c,e,f; Extended data Fig. 2e,f).

No enrichment for GO biological processes was found by transcriptomic analysis in FAT4 hCOs (Fig. 2e).

5. SCN3A: The choice of this sodium channel requires clarification. The current results do not convincingly demonstrate SCN3A involvement. Is the antibody against SCN3A validated? Antibodies for sodium channels are often nonspecific. Why were experiments conducted with Lamotrigine, which blocks all sodium channels? Showing a greater effect of Lamotrigine on mutant lines compared to controls would provide insights into sodium channel expression levels, even if not specific to SCN3A.

We appreciate the reviewer for providing this important comment, one possible explanation for the intracellular patch-clamp recordings results is an increased density of the VGSCs at the somatic level of the DCHS1 neurons. This hypothesis was supported by our transcriptomic data where several VGSCs were upregulated in DCHS1 neurons (Extended data Fig. 2d) including SCN3A and SCN1A. We tested this hypothesis exemplarily for SCN3A as it represents the most highly expressed VGSCs in the human brain and has been associated with epilepsy, intellectual disability and CM. However, following the suggestion of reviewer number 1, we have also investigated alterations in SCN8A and we found a significant increase of this VGSC isoform only in the DCHS1 mutant neurons but not in the KO lines (Extended data Fig. 3r,s).

Yet, we cannot exclude the involvement of other VGSC isoforms, and unfortunately, we cannot exclude that the SCN3A antibody may bind to other VGSC isoforms.

Regarding the lamotrigine experiment, we think that the rationale for doing it was not clearly described. As the reviewer mentioned, lamotrigine is a non-specific sodium channel blocker, that we decided to test on DCHS1 hCOs because PH patients often present with drug-resistant epileptic seizures (Battaglia et al. 1997, Dubeau et al. 1995)^{24 25}.

Unfortunately, due to the difficulty of generating new control hCOs and culture them for 9 months to be able to perform silicon probe recordings, we could not test for its effect on control hCOs where, anyway, we would still expect a comparable activity reduction.

6. Discussion: The limitations of the study should be discussed in more detail.

Following the suggestion, the limitations of the study were included in the discussion

7. Introduction: Citations 5 and 6 do not reference these gene mutations and require different citations.

We thank the reviewer for this comment, we have now updated the references

Reviewer #3 (Remarks to the Author):

In this article, the authors investigate neuronal networks in periventricular heterotopia using *FAT4* and *DCHS1* mutant iPSC-derived cerebral organoid models. The results indicate distinct electrophysiological properties between the two PH models; *DCHS1* organoids being more excitable and presenting with increased density of voltage-gated sodium channels than *FAT4* organoids. The experiments were thorough and well-carried out, identifying differences between the two mutational conditions in periventricular heterotopia in diverse assays including cell type enrichment, electrophysiology, multi-omics, and neuron morphology.

An area of improvement for this manuscript is the relevance of these models, and their distinctions, to the clinical manifestations of PH. This is briefly addressed (line 116) by indicating clinical expression is the same across the two mutations. However, this is not reflected in the findings, which report notable differences in the phenotypic manifestations of these mutations *in vitro*. To improve impact, consider correlating findings to clinical presentations of PH.

We thank the reviewer for the comment, it is always quite difficult to correlate findings *in vitro* with *in vivo* phenotypes, however we have implemented our discussion including this consideration. The reviewer will find the following text in the discussion:

“The number of described individuals with *FAT4* and *DCHS1* remains low and therefore identifying reliable clinical correlates that differentiate the two conditions remains difficult. The degree of intellectual disability and the distribution and extent of PH are broad for both conditions (Mansour et al. 2012)⁴¹.”

Reviewer #4 (Remarks to the Author):
